# Causal Inference with Treatment Measurement Error:
# A Nonparametric Instrumental Variable Approach

**Yuchen Zhu**[1]     **Limor Gultchin**[2,3]     **Arthur Gretton**[1]     **Matt Kusner**[1]     **Ricardo Silva**[1]

[1]Department of Computer Science, University College London, UK
[2]Department of Computer Science, University of Oxford, UK
[3]The Alan Turing Institute, London, UK

## Abstract

We propose a kernel-based nonparametric estimator for the causal effect when the cause is corrupted by error. We do so by generalizing estimation in the instrumental variable setting. Despite significant work on regression with measurement error, additionally handling unobserved confounding in the continuous setting is non-trivial: we have seen little prior work. As a by-product of our investigation, we clarify a connection between mean embeddings and characteristic functions, and how learning one simultaneously allows one to learn the other. This opens the way for kernel method research to leverage existing results in characteristic function estimation. Finally, we empirically show that our proposed method, MEKIV, improves over baselines and is robust under changes in the strength of measurement error and to the type of error distributions.

## 1 INTRODUCTION

Real world data poses many problems for causal effect estimation. Unmeasured confounding, the existence of hidden common causes of a treatment $X$ and an outcome of interest $Y$, is a problem that lies at the heart of many applied sciences. Solving this problem led to a variety of approaches, the most common based on the idea of instrumental variables (IVs): an auxiliary variable $Z$ independent of $Y$ upon a perfect intervention on $X$ [Pearl, 2009, Hernán and Robins, 2020], which is predictive of $X$ but not caused by it.

A less commonly studied challenge is when *the treatment is not directly observed*. For instance, we may want to learn the effect of taking a drug ($X = 1$) against not taking it ($X = 0$), where we incentivize the patients to take it or not ($Z = 1$ vs $Z = 0$). It is not necessarily the case that $X = Z$, because the patients do it at home instead

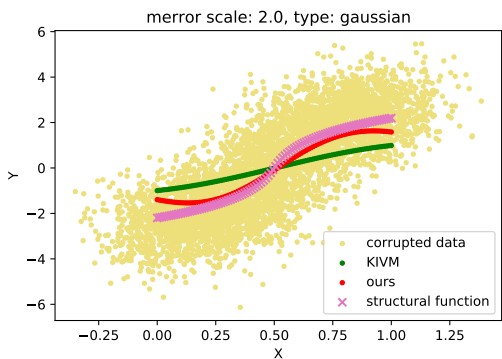

Figure 1: Comparison of curves fitted by our method and KIVM under a corrupted treatment measurement $X$ against with true curve. KIVM is a method we will discuss in the sequel, which ignores that measurements of $X$ are corrupted by additive noise.

of a hospital with supervision, and so they may not comply with the incentive. This *non-compliance* problem is compounded with the *measurement error* problem: a self-reported measurement of taking ($M = 1$) or not taking ($M = 0$) the drug does not imply $X = M$, because the patient may be lying or just forgetful. An instrumental variable approach to estimate an *average treatment effect* (ATE) such as $\mathbb{E}[Y \mid do(X = 1)] - \mathbb{E}[Y \mid do(X = 0)]$ [Pearl, 2009] may fail to give reliable results if our data consists of records of $(Z, M, Y)$, but the assumption $X = M$ does not hold.

A related issue happens when postulating latent *constructs* as causes. In a widespread example by [Bollen, 1989], a model for the effects of "industrialization level" of a country in its political freedom $Y$ is considered. We may operationalize this construct by postulating a space of possible interventions $Z$ on industrialization $X$ that keep the relation between $X$ and $Y$ invariant. However, it remains the case that $X$ is not directly observable but for indirect measurements $M$, such as the GDP or the proportion of labor force working in industry.

*Accepted for the 38[th] Conference on Uncertainty in Artificial Intelligence* (UAI 2022).

Of relevance, in both classes of indirect treatment measurement problem, is that the causal relation $\mathbb{E}[Y \mid do(x)]$ is considered to be fundamental, with $\mathbb{E}[Y \mid do(m)]$ being either zero, or poorly defined, or of secondary interest (for instance, *redefining* GDP may as well have a genuine causal impact, but this intervention is not the motivation behind understanding the causal impact of industrialization levels). In particular, measurement mechanisms may change more easily than the relation between the putative cause and the outcome of interest (we may redefine GDP, or collect data where the phrasing and timing of our questioning of a patient's compliance varies in different communities, while assuming that the relation between $X$ and $Y$ is invariant). In a way, this measurement problem is a counterpart to why estimating intention-to-treat effects, i.e. $\mathbb{E}[Y \mid do(z)]$, is not in many cases the goal of an IV analysis, despite the policy-making implications.

The need to understand effects of the mismeasured quantities on other quantities of interest motivates the study of measurement error modeling [Carroll et al., 2006, Schennach, 2016, Hernán and Robins, 2020]. Famously, even in the linear (noncausal) regression case, naïvely regressing $Y$ on a noisy measurement of $X$ results in *attenuation error*, which essentially means that the regression coefficient will be underestimated due to the measurement error [Carroll et al., 2006]. An analogous phenomenon will take place when estimating causal effects. Figure 1 depicts a kernel method that attempts to estimate a $X$-$Y$ dose-response curve, ignoring measurement error in $X$, compared against the curve found by the method we propose.

The nonlinear and confounded setting is an open domain to be explored. Schennach [2016] suggests that, in general, three measurements are needed to identify the full joint distribution of the measurements and the latent variable. However, in cases where we can make some assumptions on the error distribution, this can be reduced. Furthermore, we are not interested in the full joint distribution with the latent variable $X$, but only the parts which we need as components of the IV regression model. To that effect, we will assume that our problem follows the Markov properties of Figure 2: we are interested in the structural function $f(x) \equiv \mathbb{E}[Y \mid do(x)]$, where observationally $Y = f(x) + \epsilon$, the error term $\epsilon$ being correlated with treatment $X$. We assume that we have access to at least two treatment measurements, $M$ and $N$, and an instrumental variable $Z$.

Our contribution is threefold:

- we propose an estimator for the structural function $f(x)$ without requiring latent variable modeling. The resulting method can be applied without restrictive assumptions in the likelihood, such as the requirement for Gaussian error terms;
- in particular, we provide a method to learn the conditional mean embedding [Muandet et al., 2017] of a

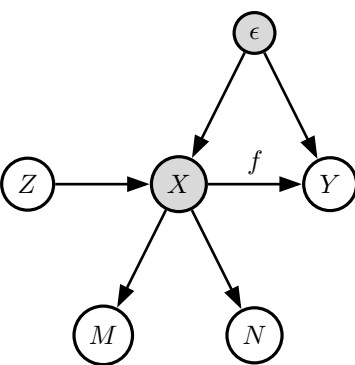

Figure 2: An instrumental variable model with confounded treatment $X$ and $Y$, where the treatment is unobservable, but has indirect measurements $M$ and $N$.

latent variable distribution, which can be applied to many two-stage IV settings;

- we propose a way to exploit the connection between characteristic function methods and kernel methods, which may be applied to many settings outside of measurement error modelling (see Section 4.1).

## 2 RELATED WORK

Measurement bias takes several forms in causal inference. See [Hernán and Robins, 2020, Chapter 9] for a textbook overview, including examples motivating different error structures. For instance, there is a growing literature on causal inference by adjusting for confounders which are only measured via proxies [e.g Kuroki and Pearl, 2014, Battistin and Chesher, 2014, Kallus et al., 2018, Miao et al., 2018, Mastouri et al., 2021, Ghassami et al., 2022]. Because different components of a causal structure contribute differently to the target estimand, it is not surprising though that conditions on identifiability in our setup vary substantively from, e.g., those required by models of confounding proxies. In general, our work is related to the large field of latent variable distribution identification, particularly those involving Markovian assumptions [Allman et al., 2009]. Even closer is the literature on *error-in-variables* regression [Carroll et al., 2006], as it provides several results for identification and estimation in the related problem of regression analysis. Schennach [2016] provides a more up-to-date review, including some comments on its use in causal effect estimation. To the best of our knowledge, no direct connection between methods for continuous causal effect estimation with both unmeasured confounding and measurement error in the treatment is described in the literature. For partially identified models in *discrete* spaces, Finkelstein et al. [2021] provide an approach based on the linear programming formulation of Balke and Pearl [1994]. It should be pointed out that the term "instrumental variable" is sometimes used in the error-in-variables regression literature as the name for a second measurement of the missing regressor $X$ [Carroll

et al., 2006, Chapter 6]. In this work, it signifies a direct cause of $X$ (the more standard definition in causal inference literature). A similar setting was studied in Gultchin et al. [2021], where $Z$ operated as a crude intervention on a complex cause $X$ and causal effects under new $Z$ interventions could be computed due to invariance. A family of methods for causal inference with corrupted data is presented by Agarwal and Singh [2021], including a linear error-in-variables formulation with Riesz representers that can be interpreted as estimating causal effects when no confounding is present. An example of effect estimation with observed confounding adjustment is Song et al. [2015], and an example combining deconfounding and measurement error correction with linear models is Vansteelandt et al. [2009]. Finally, measurement error has also been considered in the causal graph discovery literature [e.g. Silva et al., 2006, Zhang et al., 2018]. In this case, all observed variables are measurements of some underlying latent causes, and the goal is to learn the causal structure among such latent variables. Parametric assumptions are usually necessary, and no nonlinear causal effect estimators are provided when latent variables are themselves confounded by further hidden common causes.

# 3  BACKGROUND

Throughout, we use capital letters (e.g. $A$) to denote a random variable on a measurable space. We denote measurable spaces by calligraphic letters ($\mathcal{A}$), with one exception: $\mathcal{P}$, which we use to denote a probability measure. We use lowercase letters to denote realizations of a random variable ($A = a$). We will use the structural causal model (SCM) formulation of Pearl [2009], where causal relationships are represented as directed acyclic graphs (DAGs). The operator $do(\cdot)$ is defined in these models to describe the process of forcing a random variable to take a particular value, which isolates its effect on downstream variables (i.e. $\mathbb{E}[Y \mid do(A = a)]$ describes the isolated effect of $A$ on $Y$).

Our goal is to estimate the average treatment effect (ATE) $\mathbb{E}[Y \mid do(X = x)]$ given the graph in Figure 2 (equivalent to the structural function $f : \mathcal{X} \longrightarrow \mathcal{Y}$). Here $X, \epsilon$ are unobserved. We only have access to an instrument $Z$, the effect $Y$, and corrupted measurements of $X$: $M$ and $N$.

## 3.1  STRUCTURAL ASSUMPTIONS ON $p(x, y \mid z)$

When the treatment $X$ is observed and an instrument $Z$ is available, the structural function is identified by

**Assumption 1**  $Y = f(X) + \epsilon$ and $\mathbb{E}[\epsilon|Z] = 0$

**Assumption 2**  $p(x|z)$ is not constant in $z$.

Under Assumptions 1 and 2, the structural function satisfies

the following equation $\mathcal{P}_{\mathcal{Z}}$-almost surely.:

$$\mathbb{E}[Y|Z] = \mathbb{E}[f(X)|Z] = \int f(x) d\mathcal{P}_{\mathcal{X}|\mathcal{Z}} \qquad (1)$$

Typical methods fall into two categories: 1) two-stage methods (Singh et al. [2019a], Hartford et al. [2017], Xu et al. [2020]): first identify the conditional distribution directly or through estimating conditional expectations of basis functions; this is followed by identifying $f$ under the identified conditional distribution of vector of conditional expected values of basis functions; 2) moment-based methods (Zhang et al. [2020], Bennett et al. [2019]): estimate $f$ using moment conditions generated by the conditional moment restriction: $\mathbb{E}[(Y - f(X))g(Z)] = 0 \ \forall g$ measurable. A practical difference between two-stage methods and moment-based methods is that two-stage methods require separate data for each stage, and the first stage does not require the outcome observations $Y$, whereas moment-based methods require data for all variables simultaneously. In this work, we seek to identify the measurement process before identifying the structural function. We thus naturally adopt the two-stage framework since the measurement process requires only the instrument and the measurements, and not the outcome labels.

## 3.2  REPRODUCING KERNEL HILBERT SPACES

For any space $\mathcal{S} \in \{\mathcal{X}, \mathcal{Y}, \mathcal{M}, \mathcal{N}, \mathcal{Z}\}$, let $k : \mathcal{S} \times \mathcal{S} \to \mathbb{R}$ be a positive definite kernel. We denote by $\phi$ its associated canonical feature map $\phi(x) = k(x, \cdot)$ for any $x \in \mathcal{S}$, and $\mathcal{H}_{\mathcal{S}}$ its corresponding Reproducing Kernel Hilbert Space (RKHS) of real-valued functions on $\mathcal{S}$. The space $\mathcal{H}_{\mathcal{S}}$ is a Hilbert space with inner product $\langle \cdot, \cdot \rangle_{\mathcal{H}_{\mathcal{S}}}$. It satisfies two important properties: (i) $k(x, \cdot) \in \mathcal{H}_{\mathcal{S}}$ for all $x \in \mathcal{S}$, (ii) the reproducing property: for all $h \in \mathcal{H}_{\mathcal{S}}$ and $x \in \mathcal{S}, h(x) = \langle h, k(x, \cdot) \rangle_{\mathcal{H}_{\mathcal{S}}}$. For any distribution $p$ on $\mathcal{S}$, $\mu_p := \int k(x, \cdot) p(x) dx$ is an element of $\mathcal{H}_{\mathcal{S}}$ and is referred to as the kernel mean embedding of $p$ (Smola et al. [2007]). Similarly, for any conditional distribution $p(x|z)$, $\mu_{\mathcal{P}_{X|z}} := \int k(x, \cdot) p(x|z) dx$ is a *conditional mean embedding* (CME) of $p(x|z)$ (Song et al. [2009, 2013]); see Muandet et al. [2017] for a review.

## 3.3  STRUCTURE LEARNING USING KERNEL MEAN EMBEDDINGS

Provided that the structural function $f$ lies in the RKHS $\mathcal{H}_{\mathcal{X}}$, then its conditional expectation under $\mathcal{P}_{X|Z}$ can be written as $\mathbb{E}[f(X)|Z] = \langle f, \mu_{\mathcal{P}_{X|z}} \rangle_{\mathcal{H}_{\mathcal{X}}}$. In Singh et al. [2019b], the conditional mean embedding is estimated by the standard regression formula using the observed samples $\{z_j, x_j\}_{j=1}^{s_1}$ before the structural function $f$ is estimated using a second-stage sample $\{\check{z}_j, \check{y}_j\}_{j=1}^{s_2}$. We present their solution here.

The CME estimator of $\mathcal{P}_{X|z}$ is estimated using the samples

$$\{z_j, x_j\}_{j=1}^{s_1}$$

$$\hat{\mu}_{\mathcal{P}_{X|z}}^{(s_1)} = \Phi_X(K_{ZZ} + s_1\hat{\lambda}I)^{-1}\Phi_Z'\phi(z) \quad (2)$$

where $\hat{\lambda}$ is the ridge regression hyperparameter chosen using the validation procedure described in [Singh et al., 2019a, App.7.4.2]. $K_{ZZ}$ denote the kernel matrix where $(K_{ZZ})_{jl} = k(z_j, z_l)$, $(\Phi_X)_{(:,j)} = \phi(x_j)$. This is precisely the adaptation of ridge regression to multi-dimensional feature spaces to the case where the number of features can be infinite. Furthermore, if we assume that the structure function lies in an RKHS, then we can learn the function $f$ in two steps of regression: first a regression to get the CME, followed by a regression from the CME to $Y$ to obtain $f$. We go ahead to make this assumption. Importantly, we stress that the purpose of this assumption is for the nonparametric modelling of $f$, and is *not* to do with the correction of measurement error.

**Assumption 3**  $f \in \mathcal{H}_{\mathcal{X}}$

Assuming $f \in \mathcal{H}_{\mathcal{X}}$, the estimated CME $\hat{\mu}_{\mathcal{P}_{X|z}}^{(s_1)}$ is used to learn the structural function $f$ by solving the empirical analogue of the following:

$$\mathbb{E}[Y|Z] = \langle f, \mu_{\mathcal{P}_{X|z}}\rangle_{\mathcal{H}_{\mathcal{X}}} \quad (3)$$

We solve for $f$ via least squares in two stages: 1. Use $\{\check{z}_j, \check{y}_j\}_{j=1}^{s_1}$ to Monte-Carlo estimate $\mathbb{E}[Y|Z]$ and $\mu_{\mathcal{P}_{X|z}}$ (call the latter $\hat{\mu}_{\mathcal{P}_{X|z}}^{(s_1)}$); 2. Use $\{\check{z}_j, \check{y}_j\}_{j=1}^{s_2}$ to estimate $f$ via

$$\hat{f}^{(s_2)}(x) = \hat{\beta}'K_{Xx} \quad (4)$$

$$\hat{\beta} = (VV' + s_2\hat{\xi}K_{XX})^{-1}V\check{y} \quad (5)$$

$$V = K_{XX}(K_{ZZ} + s_1\hat{\lambda}I)^{-1}K_{Z\check{Z}} \quad (6)$$

where $\hat{\xi}$ is a hyperparameter. Note that $\hat{\mu}_{\mathcal{P}_{X|z}}^{(s_1)}$ enters in eq. (6): $V_{jl} = \phi(x_j)^T\Phi_X(K_{ZZ} + s_1\hat{\lambda}I)^{-1}\Phi_Z'\phi(\check{z}_l) = \hat{\mu}_{\mathcal{P}_{X|\check{z}_l}}^{(s_1)}(x_j)$. We refer the readers to Singh et al. [2019b] for the full derivation and for tuning $\hat{\xi}$.

This approach works when we observe treatment $X$. When $X$ is unobserved, this is not possible. Thus, we propose a method to learn the CME directly from corrupted measurements of $X$; then, $f$ is yielded as a mapping from the learnt CME to $Y$ as in Eq. (4) to Eq. (6). Our method is detailed in Section 4. We note that the solution of Singh et al. [2019b] requires standard conditions for kernel causal learning, which we inherit. For clarity of presentation, we detail them in the Section 3 of the Supplementary Materials.

### 3.4 CHARACTERISTIC FUNCTION IDENTIFICATION OF A LATENT VARIABLE USING MISMEASURED OBSERVATIONS

The main obstacle in the learning of the CME $\mu_{\mathcal{X}|\mathcal{Z}}$ is the lack of observed data of $X$. To this end, we first review a strongly related problem, which is to identify the characteristic function of $p(x|z)$ using corrupted observations $M$ and $N$. The following assumptions are needed.

**Assumption 4**  Measurement errors enter additively:

$$M = X + \Delta M \quad (7)$$

$$N = X + \Delta N \quad (8)$$

**Assumption 5**  The measurement errors are uncorrelated with each other, $\Delta M$ is uncorrelated with $X$, $\Delta N$ is independent with $X$, and $\epsilon$ is uncorrelated with $\Delta N$:

$$\mathbb{E}[\Delta M|X, \Delta N] = 0 \quad (9)$$

$$X \perp\!\!\!\perp \Delta N \quad (10)$$

$$\mathbb{E}[\epsilon|\Delta N] = 0 \quad (11)$$

As $X$ is unobserved and can be redefined up to any invertible transformation, Eq. (7) is not imposing further constraints besides a monotonic relation between $M$ and $X$ in expectation. Eqs. (9) and (11) are weaker formulations of conditional independence statements $\Delta M \perp\!\!\!\perp \{X, \Delta N\}$ and $\epsilon \perp\!\!\!\perp \Delta N$.

**Remark 1**  Eq. (8) is a restrictive assumption. However, we point out that it can be relaxed and the relaxed setting can be reduced to our setting. Thus we focus on the simplified setting where future methods can extend from; we discuss one way to relax the assumption in Section 2 of the Supplementary Materials.

With two measurements, Schennach [2004] provides a constructive estimator for the moments of latent variables. Our work uses a special case of their theorem which we state below.

**Assumption 6**  $\mathbb{E}[|X|] < \infty$ and $\mathbb{E}[|\Delta M|] < \infty$

**Corollary 1**  Given Assumptions 4-6, the characteristic function of $X$ given $Z = z$, i.e. $\psi_{\mathcal{P}_{X|z}}(\alpha)$, is equal to

$$\overbrace{\mathbb{E}_{\mathcal{P}_{X|z}}[e^{i\alpha X}]}^{\psi_{\mathcal{P}_{X|z}}(\alpha):=} = \exp\left(\int_0^\alpha i\frac{\mathbb{E}[Me^{i\nu N}|z]}{\mathbb{E}[e^{i\nu N}|z]}d\nu\right). \quad (12)$$

*Proof.* Follows directly from [Schennach, 2004, Theorem 1], where the original phrased the equality for the marginal distribution $p(x, m, n)$. $\qquad\square$

Since characteristic functions are exact representations of probability distributions, Corollary 1 says that we may model $\mathcal{P}_{X|z}$ through modelling $\mathcal{P}_{M,N|z}$ and $\mathcal{P}_{N|z}$, and the mathematical relation is specified by Eq. (12).

## 4   METHOD

In this section we will show how to recover $\mathbb{E}[Y \mid do(X = x)]$. To do so, recall that all we need is the kernel mean

embedding $\mu_{\mathcal{P}_{X|Z}}$. We begin by demonstrating that estimating $\mu_{\mathcal{P}_{X|Z}}$ boils down to estimating the characteristic function $\psi_{\mathcal{P}_{X|Z}}$. We then introduce a trick for solving integral equations that we call the *differentiation trick* which allows us to estimate $\psi_{\mathcal{P}_{X|Z}}$ without explicitly estimating the integral. Finally, we give a full procedure for estimating $\mathbb{E}[Y \mid do(X=x)]$ and describe advantages of our approach.

## 4.1 FROM KERNEL MEAN EMBEDDINGS TO CHARACTERISTIC FUNCTIONS

For simplicity, we limit our description to $\mathbb{R}$. However, all of the following arguments can be extended trivially to $\mathbb{R}^d, d > 1$. First recall the Fourier transform:

$$\tilde{h}(\alpha) = \frac{1}{2\pi}\int_{-\infty}^{\infty} h(x)e^{-i\alpha x}dx$$

and the inverse Fourier transform:

$$h(x) = \int_{-\infty}^{\infty} \tilde{h}(\alpha)e^{i\alpha x}d\alpha.$$

Further, we assume the following.

**Assumption 7** (Symmetric, characteristic and translation-invariant kernels) $k(x, \cdot), k(m, \cdot), k(n, \cdot)$ are symmetric, characteristic and translation-invariant kernels.

Kernel symmetry is a standard assumption in ML as kernel functions are generally real. Characteristic kernels allow us to embed probability distributions uniquely in an RKHS. Translation-invariant kernels allow us to consider the probability measure associated with kernel functions.

Under Assumption 7, we can write $k(x, y) = k(x - y)$ and $k(t)$ is positive definite. By Bochner's theorem, we know that $k$ can be written as the Fourier transform of a unique measure $\tilde{k}$:

$$k(t) = \frac{1}{2\pi}\int_{-\infty}^{\infty} e^{-i\alpha t}\tilde{k}(\alpha)d\alpha$$

$$\text{i.e. } k(x, y) = \int_{-\infty}^{\infty} e^{-i\alpha(x-y)}q(\alpha)d\alpha$$

where $q(\alpha) := \frac{1}{2\pi}\tilde{k}(\alpha)$. As illustrated in e.g. Fukumizu [2008], we may construct an RKHS on the entire real line using Fourier transforms as feature maps:

$$\mathcal{H}_X = \left\{ f \in \mathcal{L}^2(\mathbb{R}, dx) \middle| \int_{-\infty}^{\infty} \frac{\left|\tilde{f}(\alpha)\right|^2}{q(\alpha)}d\alpha < \infty \right\}$$

$$\langle f, g \rangle_{\mathcal{H}_X} = \int_{-\infty}^{\infty} \frac{\tilde{f}(\alpha)\overline{\tilde{g}(\alpha)}}{q(\alpha)}d\alpha$$

Now consider the Fourier transform of $k(x, \cdot)$, where $x$ is fixed. Since we know that $k(x, y) = \int e^{-i\alpha x}e^{i\alpha y}q(\alpha)d\alpha$,

by inspection we realise $\tilde{k}(x, \alpha) = e^{-i\alpha x}q(\alpha)$, recovering the identity that $k(x, y) = \int_{-\infty}^{\infty} \frac{e^{-i\alpha x}q(\alpha)e^{i\alpha y}q(\alpha)}{q(\alpha)}d\alpha = \langle k(x, \cdot), k(y, \cdot)\rangle_{\mathcal{H}_X}$.

Recall the definition of the conditional mean embedding of $\mathcal{P}_{X|z}$ for a particular $z$: $\mu_{\mathcal{P}_{X|z}}(y) := \int k(x, y)p(x|z)dx$. When all variables are observed, the conditional mean embedding (CME) can be estimated by samples $\{x_j, z_j\}_{j=1}^s$:

$$\hat{\mu}_{X|z}^{(s)}(y) = \sum_{j=1}^{s} \hat{\gamma}_j^{(s)}(z)k(x_j, y) \tag{13}$$

where

$$\hat{\gamma}_j^{(s)}(z) = (K_{ZZ} + s\hat{\lambda}^{(s)}I)^{-1}K_{Zz} \tag{14}$$

Taking the Fourier transform of $\hat{\mu}_{X|z}^{(s)}(y)$:

$$\tilde{\hat{\mu}}_{X|z}^{(s)}(\alpha) = \sum_{j=1}^{s} \hat{\gamma}_j^{(s)}(z)e^{-i\alpha x_j}q(\alpha)$$

$$= q(\alpha)\underbrace{\sum_{j=1}^{s} \hat{\gamma}_j^{(s)}(z)e^{-j\alpha x_j}}_{=:\hat{\psi}_{\mathcal{P}_{X|z}}^{(s)}(-\alpha)}$$

Define the $s-$sample estimate of the characteristic function $\hat{\psi}_{\mathcal{P}_{X|z}}^{(s)}(\alpha) := \sum_{j=1}^{s} \hat{\gamma}_j^{(s)}(z)e^{i\alpha x_j}$ with $\{x_j\}_{j=1}^s \sim \mathcal{P}_{X|z}$. Next, we show that $\hat{\psi}_{\mathcal{P}_{X|z}}^{(s)} \longrightarrow \psi_{\mathcal{P}_{X|z}}$ in $\mathcal{L}^2(\mathbb{R}, q)$ if and only if $\hat{\mu}_{\mathcal{P}_{X|z}}^{(s)} \longrightarrow \mu_{\mathcal{P}_{X|z}}$ in $\mathcal{H}_X$.

**Theorem 1** (Convergence in CME is identical to convergence in characteristic function) . Let $k : \mathcal{X} \times \mathcal{X} \longrightarrow \mathbb{R}$ be a symmetric, positive definite, and translationally invariant characteristic kernel, then for a (conditional) probability measure on $\mathcal{X}$, denoted $\mathcal{P}_{X|z}$, we have that $\hat{\psi}_{\mathcal{P}_{X|z}}^{(s)} \longrightarrow \psi_{\mathcal{P}_{X|z}}$ in $\mathcal{L}^2(\mathbb{R}, q)$ if and only if $\hat{\mu}_{\mathcal{P}_{X|z}}^{(s)} \longrightarrow \mu_{\mathcal{P}_{X|z}}$ in $\mathcal{H}_{\mathcal{X}}$. Moreover, whenever either converges, the other converges at the same rate.

We provide the proof in Section 6 of the Supplementary Materials.

This means learning the characteristic function $\psi_{\mathcal{P}_{X|z}}$ in $\mathcal{L}^2(\mathbb{R}, q)$ simultaneously gives us a precise estimate of the kernel mean embedding $\mu_{\mathcal{P}_{X|z}}$ in $\mathcal{H}_X$.

## 4.2 LEARNING THE LATENT CHARACTERISTIC FUNCTION

We now show how to learn the latent characteristic function which will give us the latent kernel mean embedding.

**Notation.** To lighten notation, from now on we will use $\hat{f}$ to denote the empirical estimate of a quantity $f$, and only use $\hat{f}^{(s)}$ when we need to be specify the sample size $s$.

**What if we are able to observe $X$?** When $X$ is observed, $\hat{\mu}_{\mathcal{P}_{X|z}}^{(s)}$ can be obtained directly and it can be shown that $\hat{\mu}_{\mathcal{P}_{X|z}}^{(s)} \longrightarrow \mu_{\mathcal{P}_{X|z}}$ as $s \longrightarrow \infty$. By Theorem 1, the same samples and $\lambda$ which closely estimate the CME $\mu_{\mathcal{P}_{X|z}}$ would also closely estimate the characteristic function $\psi_{\mathcal{P}_{X|z}}$, and vice versa [1]. Thus, when $s$ is suitably large, we can accurately approximate the right hand side of (12) as

$$\exp\left(\int_0^\alpha i \frac{\mathbb{E}[Me^{i\nu N}|z]}{\mathbb{E}[e^{i\nu N}|z]}d\nu\right) \approx \sum_{j=1}^s \hat{\gamma}_j(z)e^{i\alpha x_j} \quad (15)$$

where $\hat{\gamma}_j(z)$ is specified by Eq. (14). Recall that this term also depends on $\hat{\lambda}$. To make this explicit we write $\hat{\gamma}_j^{\hat{\lambda}}(z)$.

**Solving for $X$.** Given eq. 15 we make the following observation: given samples of $\{z_j\}_j$, the estimate $\hat{\psi}_{\mathcal{P}_{X|z}}$ only depends on $\{x_j\}_j$ and $\hat{\lambda}$.

Therefore, we can solve for $\{x_j\}_j, \hat{\lambda}$ by minimising the discrepancy between both sides of Eq. (15) over $\{x_j\}_j, \hat{\lambda}$:

$$\{\hat{x}_j\}_j, \hat{\lambda}_X = \operatorname*{argmin}_{\{x_j\}_j, \hat{\lambda}} \mathbb{E}_{q(\alpha), \mathcal{P}_{\check{Z}}}\left[\left(\sum_{j=1}^s \hat{\gamma}_j^{\hat{\lambda}}(\check{Z})e^{i\alpha\hat{x}_j} - \eta\right)^2\right]$$
$$\text{(16)}$$
$$\text{with } \eta = \exp\int_0^\alpha \left(i\frac{\mathbb{E}[Me^{i\nu N}|\check{Z}]}{\mathbb{E}[e^{i\nu N}|\check{Z}]}d\nu\right)$$

The expectation is taken over $q(\alpha)$ and $\mathcal{P}_{\check{Z}}$ because, had the $X$−samples been observed, the convergence of characteristic function is in $\mathcal{L}^2(\mathbb{R}, q)$ and the $\check{Z}$ distribution does not have to equal to the one used to learn the CME, as long as the two have the same support. To estimate $\eta$ requires two components of approximation: a) finite-sample approximation of $\mathbb{E}[e^{i\nu N}|\check{Z}]$ and $\mathbb{E}[Me^{i\nu N}|\check{Z}]$, b) computation of the integral $\int_0^\alpha \left(i\frac{\mathbb{E}[Me^{i\nu N}|\check{Z}]}{\mathbb{E}[e^{i\nu N}|\check{Z}]}d\nu\right)$, given a). While it is possible to use numerical methods such as quadrature to approximate the integral, we propose to save the second component by differentiation.

**The Differentiation Trick.** We now describe a trick for handling intractable integrals when solving a system of equations. First, let us reproduce eq. (12) below

$$\overbrace{\underbrace{\mathbb{E}_{\mathcal{P}_{X|z}}[e^{i\alpha X}]}_{\psi_{\mathcal{P}_{X|z}}(\alpha):=}} = \exp\left(\int_0^\alpha i\frac{\mathbb{E}[Me^{i\nu N}|z]}{\mathbb{E}[e^{i\nu N}|z]}d\nu\right).$$

We can take the natural logarithm and differentiate both

---

[1]This should be possible for $z$ from an unseen distribution $\mathcal{P}_{\check{z}}$ provided the unseen distribution has the same support as the training distribution $\mathcal{P}_Z$.

---

sides of eq. (12), and substitute the samples of $\check{\mathcal{Z}}$:

$$\frac{\mathbb{E}[Xe^{i\alpha X}|\check{z}]}{\mathbb{E}[e^{i\alpha X}|\check{z}]} = \frac{\mathbb{E}[Me^{i\alpha N}|\check{z}]}{\mathbb{E}[e^{i\alpha N}|\check{z}]} \quad (17)$$

Since differentiation is a many-to-1 operation, we need to verify that the solution to Eq. (17) is also the solution to Eq. (12).

**Lemma 1** Considering differentiable functions $\mathbb{C}^n \to \mathbb{C}$. Denote $f'(x) := \frac{d}{dx}f(x)$. Then if $f' = g'$ and $f(a) = g(a) = b, a, b \in \mathbb{C}$, then $f = g$.

*Proof.* If $f' = g'$, then $f = g + C$ for some $C \in \mathbb{C}$. But $f(a) - g(a) = b - b = 0$, so $C = 0$. $\square$

**Theorem 2** The (conditional) distribution of $X$, denoted by $\mathcal{P}_{X|z}$, which satisfies Eq. (17) is unique, and therefore is the same as the solution to (12).

The proof relies on the fact that characteristic functions are always 1 at $\alpha = 0$ (Section 6 of the Supplementary Materials).

**When should one use the differentiation trick?** When estimation for the target function/parameter requires evaluating an intractable integral, one can think of using the differentiation trick. Lemma 1 specifies one condition where this can be done. Note that there are more situations where the differentiation trick can be applied, such as when all functions in the target class have the same normalization constant. We summarize two situations where the differentiation trick can be applied:

- When the target function class is itself normalized, or fixed at certain input values. Examples of this which may be of interest to machine learning practitioners are: a) probability densities, which always integrates to 1, b) cumulative distributions, which is always 1 at $\infty$.

- When an invertible transformation of the function class is normalized or fixed at certain inputs. In those cases, one can in principle solve the problem in the normalized function class, and then apply the invertible transformation to go back to original class.

**Towards a sample-based estimator.** As discussed, we may replace $\mathbb{E}[e^{i\alpha X}|\check{z}]$ and $\mathbb{E}[e^{i\alpha N}|\check{z}]$ with their finite-sample estimates $\hat{\psi}_{\mathcal{P}_{X|\check{z}}}$ and $\hat{\psi}_{\mathcal{P}_{N|\check{z}}}$. For $\mathbb{E}[Xe^{i\alpha X}|\check{z}]$ and $\mathbb{E}[Me^{i\alpha N}|\check{z}]$, we realise that $\mathbb{E}[Xe^{i\alpha X}|\check{z}] = \frac{\partial}{\partial\alpha}\mathbb{E}[e^{i\alpha X}|\check{z}]$, and $\mathbb{E}[Me^{i\alpha N}|\check{z}] = \frac{\partial}{\partial v}\Big|_{v=0} \mathbb{E}[e^{i(\alpha N + vM)}|\check{z}]$. Thus, we replace them with $\frac{\partial}{\partial\alpha}\hat{\psi}_{\mathcal{P}_{X|z}}(\alpha)$ and $\frac{\partial}{\partial v}\Big|_{v=0} \hat{\psi}_{\mathcal{P}_{M,N|z}}(\alpha, v)$ respectively. The full expressions of $s$−sample estimates for $\hat{\psi}_{\mathcal{P}_{X|z}}(\alpha)$, $\hat{\psi}_{\mathcal{P}_{N|z}}(\alpha)$, $\hat{\psi}_{\mathcal{P}_{M,N|z}}(v, \alpha)$ and the relevant derivatives are stated in Section 4 of the Supplementary Materials.

Therefore, we arrive at the new objective function:

$$\{\hat{x}_j\}_{j=1}^s, \hat{\lambda}_X =$$
$$\underset{\{x_j\}_{j=1}^s, \hat{\lambda}_X}{\operatorname{argmin}} \ \mathbb{E}_{q(\alpha), \mathcal{P}_{\check{Z}}} \left[ \left( w_X(\alpha, \check{Z}) - w_{MN}(\alpha, \check{Z}) \right)^2 \right]$$
(18)

with
$$w_X(\alpha, \check{Z}) = \frac{\sum_{j=1}^s x_j \hat{\gamma}_X(\check{Z})_j e^{i\alpha x_j}}{\sum_{j=1}^s \hat{\gamma}_X(\check{Z})_j e^{i\alpha x_j}}$$
(19)

$$w_{MN}(\alpha, \check{Z}) = \frac{\sum_{j=1}^s m_j \hat{\gamma}_{M,N}(\check{Z})_j e^{i\alpha n_j}}{\sum_{j=1}^s \hat{\gamma}_N(\check{Z})_j e^{i\alpha n_j}}$$
(20)

$w_{MN}$ is the sample estimate for the integrand in $\eta$ from Eq. (16). We can interpret the output values of $w_{MN}$ as the labels for the supervised learning task defined by Eq. (18), the $(\alpha, \check{z})$ as inputs, and the $\{x_j\}$ and $\hat{\lambda}_X$ are the parameters. As soon as we have obtained the optimal $\{\hat{x}_j\}_{j=1}^s$ and $\hat{\lambda}_X$, we can substitute into Eq. (13) and Eq. (14) to obtain the CME estimate $\hat{\mu}_{\mathcal{P}_{X|z}}$.

## 4.3 ALGORITHM

We propose *MEKIV*: **M**easurement-**E**rror-corrected **K**ernel **I**nstrumental **V**ariable regression. Two independent samples are needed: $\{z_j, m_j, n_j\}_{j=1}^{s_1}$ and $\{\check{z}_j, \check{y}_j\}_{j=1}^{s_2}$.

Thanks to Theorem 1, In step 1 of the MEKIV, we use $\{z_j, m_j, n_j\}_{j=1}^{s_1}$ to compute the sample estimates of the conditional kernel mean embeddings of $\mathcal{P}_{N|z}$ and $\mathcal{P}_{M,N|z}$, which in large sample size is guaranteed to converge to the ground truth Singh et al. [2019b]. By Theorem 1, this also gives us a sample estimate of the characteristic functions which converges in $\mathcal{L}^2$ of their measures induced by their respective kernels.

Step 2 of the MEKIV learns the characteristic function of $\mathcal{P}_{X|Z}$ by optimising for the $X$ samples using the training objective in Eq. (18). Again by Theorem 1, a good estimate of the characteristic function gives us a good estimate of the conditional kernel mean embedding.

In Step 3, MEKIV uses the learnt kernel conditional mean embedding and the second samples $\{\check{z}_j, \check{y}_j\}_{j=1}^{s_2}$ to estimate the structural function $f$ - equivalent to the stage 2 of the KIV (Singh et al. [2019b]).

The pseudocode of our complete algorithm can be found in Algorithm 1 and 2 in the Supplementary Materials.

**Step 1.** From the first sample $\{z_j, m_j, n_j\}_{j=1}^{s_1}$, learn the conditional mean embedding of $p(m|z)$ and $p(m,n|z)$ using the result stated in Eq. (2), Section 3.3:

$$\hat{\mu}_{\mathcal{P}_{N|z}}^{(s_1)}(\cdot) = \sum_{j=1}^{s_1} (\hat{\gamma}_N^{(s_1)}(z))_j k(n_j, \cdot),$$
(21)

with
$$\hat{\gamma}_N^{(s_1)}(z) = (K_{ZZ} + s_1 \hat{\lambda}_N I)^{-1} K_{Zz}$$
(22)

Similarly, it can be shown that:

$$\hat{\mu}_{\mathcal{P}_{M,N|z}}^{(s_1)}(\cdot) = \sum_{j=1}^{s_1} (\hat{\gamma}_{M,N}(z))_j k((m_j, n_j), \cdot), \quad (23)$$

where
$$\hat{\gamma}_{M,N}^{(s_1)}(z) = (K_{ZZ} + s_1 \hat{\lambda}_{M,N} I)^{-1} K_{Zz}$$
(24)

**Remark 2** (23) allows the use of product kernels.

**Step 2.** After obtaining from Step 1 the quantities: $\hat{\gamma}_N$ and $\hat{\gamma}_{MN}$, Step 2 creates samples $\{\alpha_j\}$, $\{\check{z}_j\}$ and $\{(w_{MN})_j\}$. To this end, Step 2 samples $\{\alpha_j\}_{j=1}^{s_2}$ from $q(\alpha)$, and uses $\{\check{z}_j\}_{j=1}^{s_2}$ unseen in Step 1. In general, $\{\check{z}_j\}_{j=1}^{s_2}$ can be drawn from any distribution $\mathcal{P}_{\check{Z}}$ with the same support as $\mathcal{P}_Z$. To maximize sample usage, we take all pairs in the cross product $\{\alpha_j\}_{j=1}^{s_2} \times \{z_j\}_{j=1}^{s_2}$, giving $(s_2)^2$ pairs: $\{\alpha_j, \check{z}_j\}_{j=1}^{(s_2)^2}$ - here we overload notation $\{\check{z}_j\}$ to be both before and after taking the cross product. We input each pair of $\{\alpha_j, \check{z}_j\}$ into Eq. (20) to generate the labels $\{(w_{MN})_j\}_{j=1}^{(s_2)^2}$. The process of sampling $\{\alpha_j\}$ from $q(\alpha)$ has a close connection with the Random Fourier Features literature (Bach [2017], Sriperumbudur and Szabó [2015], Rahimi and Recht [2007]).

We now seek $\{x_j\}_{j=1}^{s_1}$ and $\hat{\lambda}_X$ in order to minimize the following objective, which is the empirical analogue of Eq. (18):

$$\{\hat{x}_j\}_{j=1}^{s_1}, \hat{\lambda}_X = \underset{\{x_j\}_{j=1}^{s_1}, \hat{\lambda}_X}{\operatorname{argmin}} \sum_{j=1}^{(s_2)^2} \left[ \left( w_X(\alpha_j, \check{z}_j) - (w_{MN})_j \right)^2 \right]$$
(25)

For clarity, Step 2 is illustrated in Algorithm 2 (see Supplementary Materials).

**Step 3.** Given estimates of $\{x_j\}_{j=1}^{s_1}$ and $\hat{\lambda}_X$, we obtain the empirical estimate $\hat{\mu}_{\mathcal{P}_{X|z}}$. Along with the samples $\{\check{z}_j, \check{y}_j\}_{j=1}^{s_2}$, we obtain the solution for $\hat{f}^{s_1}$. The procedure is identical to the Stage 2 estimation of KIV Singh et al. [2019b], for which we stated the derived estimator in Section 3.3. Our solution for $f$ is:

$$\hat{f}^{(s_2)}(x) = (\hat{\beta})' K_{\hat{X}x}$$
(26)

with
$$\hat{\beta} = (VV' + s_2 \hat{\xi} K_{\hat{X}\hat{X}})^{-1} V \check{y}$$
(27)

$$V = K_{\hat{X}\hat{X}} (K_{ZZ} + s_1 \hat{\lambda} I)^{-1} K_{Z\check{Z}}$$
(28)

## 4.4 ADVANTAGES OF MEKIV

We highlight the benefits of MEKIV:

- MEKIV is **free of distributional assumptions**: as long as the measurement error satisfies the mean-independence conditions in Eq. (9)-(11), the distributions can have any shape.

- **Computational efficiency**: MEKIV models only the CME of $\mathcal{P}_{X|Z}$, and in particular, no modelling of the full joint distribution $\mathcal{P}_{X|Z,M,N}$ as is commonly done in standard latent variable modelling.

- **Ease of implementation**: Unlike standard latent variable modelling, which is typically hard to train due to the large number of hyperparameters, MEKIV is easy to implement and works stably without large efforts in tuning.

# 5 EXPERIMENTS

In this section we evaluate the empirical performance of MEKIV across multiple designs and against baselines. In particular, we compare to three baselines: A) KernelIV Singh et al. [2019b] with ground truth X provided from an oracle (KIV-Oracle); B) KernelIV taking $M$ as the true treatment observations (KIV-M); C) since taking the average of independent errors reduces the error variance, we also compare with KernelIV taking $(M + N)/2$ as the true treatment observations (KIV-MN).

We run each estimator on three designs. The *linear* design Chen and Christensen [2017] involves learning the structural function $f(x) = 4x - 2$, where $X$ is unseen and we are only given corrupted measurements of treatment $(M, N)$, continuous instrument $Z$, and observations of outcome variables $Y$ which is confounded with the true treatments $X$. The *sigmoid* design Chen and Christensen [2017] involves learning the structural $f(x) = \ln(|16x - 8| + 1) \cdot sgn(x - 0.5)$ under the same data generating process otherwise. The *demand* design Hartford et al. [2017] involves learning demand function $h(p, t, s) = 100 + (10 + p) \cdot s \cdot \psi(t) - 2p$ where $\psi(t)$ is a complicated nonlinear function. A data tuple including the ground truth treatments consists of $(Y, P, T, S, C)$ where $Y$ is sales, $P$ is price, $T$ is time of year, $S$ is customer sentiment (a discrete variable), and $C$ parameterizes the supply cost shift. A parameter $\rho \in \{0.25, 0.5, 0.9\}$ calibrates the confounding level of $P$ by supply-side market forces. We set $X := (P, T, S)$ and instruments are $Z := (C, T, S)$.

Since the originally proposed design is one where $X$ is observed, we construct $M$ and $N$ from $X$ and we mask $X$ from all algorithms except KIV-Oracle. For the demand design where $X$ is 3-dimensional, we mask only the dimension corresponding to $P$. For each design we construct $M$ and $N$ from adding noise on $X$. We analyze the robustness of MEKIV in two dimensions. First, we vary the measurement error distribution: we implement a *Gaussian* additive noise design and a multi-modal *Mixture of Gaussian* additive noise design where we mix two Gaussian distributions, centred at twice the standard deviation of $X$ away from 0 on either side. Second, for each measurement error distribution, we vary their standard deviation. For both designs, we set the standard deviation of the Gaussian(s) to be $\{0.5, 1, 2\}$ times the standard deviation of the ground truth $X$.

For the linear and sigmoid design, we implement 30 simulations for each algorithm, measurement error distribution (merror type) and measurement error standard deviation (merror level). For the demand dataset, due to time constraints, we implement 30 simulations for the Mixture of Gaussian measurement error distribution and 10 for the Gaussian distribution, for each algorithm and measurement error standard deviation. We calculate MSE with respect to the true structural function $f$. Figure 3, 4 and Figure 6 (Supplementary Materials) plots the results in each design, measurement error distribution type, and measurement error level. We expect KIV-Oracle to be the best across all methods and its performance is viewed as an upper bound for the other algorithms. MEKIV beats all other baselines in the highest measurement error level setting and is robust to non-classical measurement error as demonstrated by its performance under Mixture of Gaussian error.

# 6 EXEMPLARY REAL WORLD SCENARIOS

**Measurement error from survey data: effect of income on children's cognitive outcome.** Dahl and Lochner [2012] investigated the impact of family income on children's development. The primary concern with using a regression-type method is that family income and children's development is confounded by other family characteristics, such as parents' cognitive ability. Dahl and Lochner thus proposed the state's Earned Income Tax Credit (EITC) scheme as an instrumental variable to correct for the confounding. Here, they exploit the fact that the EITC scheme expands over the years, so, via this, they can capture the variation of total family income independent from that caused by intrinsic family characteristics. They base their analysis on the panel dataset in the Children of the National Longitudinal Survey of Youth. Survey data are known to contain measurement errors (Carroll et al. [2006]). Moreover, the family income measured by the survey in two consecutive years can be posed as repeated measurements of true total family income - assuming that family income is a stable variable that does not vary drastically over a short number of years. Moreover, Dahl and Lochner [2012] took a linear model approach for simplicity, but our method can be used for a nonlinear analysis. We apply our method to this dataset and discuss the experiment in detail in Supplementary Materials.

In conclusion, we find that EITC as an instrument is weak, so we prescribe that for a meaningful analysis of the hypothesis of income-on-children's-outcome, a stronger perturbation on the income is required. For example, this can be done by selecting a neighbourhood for which the strength of EITC parameters is increased for some number of years.

## 6.1 UNDERSTANDING HOW STUDENT SKILLS IMPACT THEIR LONG-TERM OUTCOME

We consider the following thought experiment: an education trial may take place such that the teachers exercise certain educational strategies to improve the skills of students, in

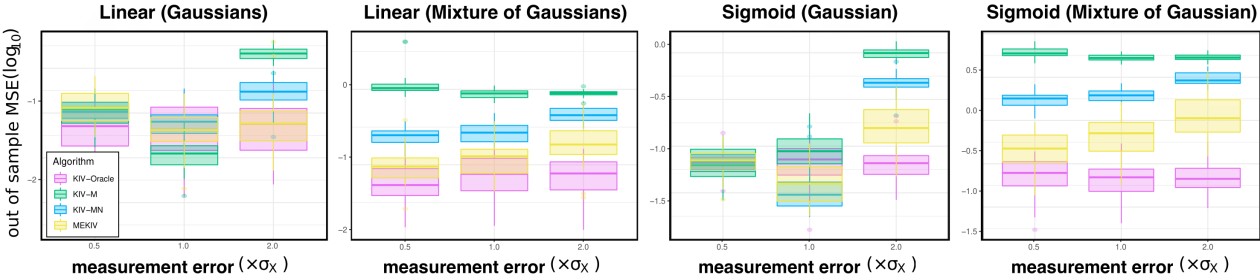

Figure 3: Out of sample MSEs ($\log_{10}$) for linear and sigmoid designs.

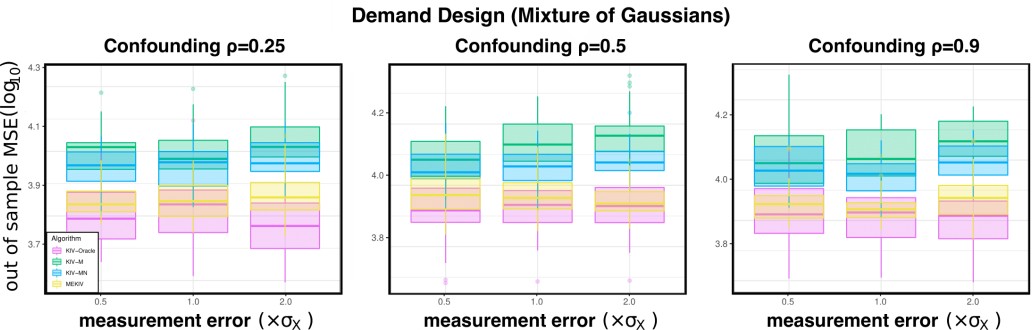

Figure 4: Out of sample MSEs ($\log_{10}$) for demand design.

the hopes of ultimately improving their long-term outcome. Such an educational strategy may be the introduction of challenging science projects; this would be $Z$. This may be hypothesized to encourage the students to understand taught concepts better and apply their knowledge to more complex scenarios. The resultant short-term changes in these skills ($X$) may be measured by test scores ($M, N$), which contain measurement error. The project should then follow the cohort of students to see what they achieve years later when they reach adulthood ($Y$). We may then run our method to determine the nonlinear causal relationship from skills, such as logical thinking and creative application of knowledge, to long-term outcomes.

## 7 CONCLUSION AND FUTURE WORK

We propose MEKIV, an instrumental variable approach for confounded structural learning when the treatment variable is measured with error. We clarify a connection between mean embedding learning and characteristic function learning, showing that the two can be done simultaneously. In constructing our algorithm, we introduce the 'differentiation trick' which allows target function recovery while avoiding the computation of an intractable integral. Our method performs well on both Gaussian and non-Gaussian measurement error, and is robust over increasing measurement error levels.

Our method should work well when interventional data is present, but when only purely observational data is present,

the instrumental variable assumption may need to be relaxed. This is due to the rarity of instrumental variables in observational studies - instrumental variables in observational studies are often weak, and in some cases they might not even be valid. We leave this for future work. Nevertheless, the ubiquity of instrumental variable assumptions suggests that our approach should be widely applicable; our proposed methodology connecting kernel learning and characteristic function learning also carries independent interest and may find applications outside of the topic of treatment effect estimation considered in this paper.

## Acknowledgements

We are grateful to Amlan Banaji and François-Xavier Briol for their insightful comments. We also thank the reviewers for the thoughtful reviews. YZ acknowledges support by the Engineering and Physical Sciences Research Council with grant number EP/S021566/1. This work was partially supported by an ONR grant number N62909-19-1-2096 to RS.

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
