# OpenReview forum: "Causal Inference with Treatment Measurement Error: A Nonparametric Instrumental Variable Approach"
_auai.org/UAI/2022/Conference — UAI 2022 Oral_

### Official Review · Reviewer_AsnF · 2022-04-05

**Q2(1) Originality/Novelty:** 3
**Q2(2) Significance/Impact:** 2
**Q2(3) Correctness/Technical Quality:** 3
**Q2(6) Clarity Of Writing:** 3
**Q6 Overall Score:** 7
**Q8 Confidence In Your Score:** 3

**Q1 Summary And Contributions:**

The authors propose a causal effect algorithm for situations where treatment is unobservable (or corrupted) but two measurements and an instrumental variable are available. The algorithm is based on kernel mean embedding methods and framed in a nonlinear SEM with additive, but potentially correlated noise, i.e. confounding.

**Q2 Assessment Of The Paper:**

More detailed information regarding each of these aspects is given below:

**Q2(4) Quality Of Experiments (Optional):**

2: Fair: The experimental evaluation is weak: important baselines are missing, or the results do not adequately support the main claims.

**Q2(5) Reproducibility:**

3: Good: Key resources (e.g., proofs, code, data) are available and key details (e.g., proofs, experimental setup) are sufficiently well-described for competent researchers to confidently reproduce the main results.

**Q3 Main Strengths:**

The paper is clearly written and well structured. The method itself seems interesting with the main advantage of being free of specific distributional assumptions.

**Q4 Main Weakness:**

Relatively small simulation section with only 10 replications and only one competitor.

**Q5 Detailed Comments To The Authors:**

Some typos:

-eq (13) $\mu_{P_X|z}$

-eq (15) dv missing


**Q7 Justification For Your Score:**

The method of learning characteristic functions to estimate the cme seems interesting with promising numerical results.

**Q9 Complying With Reviewing Instructions:**

1: Yes.

---

### Official Review · Reviewer_NC43 · 2022-04-10

**Q2(1) Originality/Novelty:** 3
**Q2(2) Significance/Impact:** 2
**Q2(3) Correctness/Technical Quality:** 3
**Q2(6) Clarity Of Writing:** 2
**Q6 Overall Score:** 7
**Q8 Confidence In Your Score:** 4

**Q1 Summary And Contributions:**

This paper proposes a kernel-based nonparametric estimator for the structural function about the causal effect when the treatment variable is measured with error and there is unmeasured confounding. The idea for estimating the structural function is new.

**Q10 Ethical Concerns (Optional):**

There is no ethical concern.

**Q2 Assessment Of The Paper:**

More detailed information regarding each of these aspects is given below:

**Q2(4) Quality Of Experiments (Optional):**

2: Fair: The experimental evaluation is weak: important baselines are missing, or the results do not adequately support the main claims.

**Q2(5) Reproducibility:**

2: Fair: Key resources (e.g., proofs, code, data) are unavailable but key details (e.g., proof sketches, experimental setup) are sufficiently well-described for an expert to confidently reproduce the main results.

**Q3 Main Strengths:**

The proposed algorithm uses the connection between characteristic function methods and kernel methods to solve the unobserved values of the treatment variable.

The introduction of the proposed method is detailed and clear.


**Q4 Main Weakness:**

The algorithm needs to solve the true unobserved values of treatment variable. If the sample size is large, the number of values to be estimated will also be very large. There is not theoretical guarantee that the algorithm will work well.

The two real examples in introduction are not very suitable, they cannot meet the measurement error models in Assumption 4.

The paper conducts some experiments, but the explanation of results is not clear. The paper says that "we  implement 10 simulations and calculate MSE with respect to the true structural function f." What is the meaning of MSE here? Figures 3, 4, and 6 show the boxplots for each method. If the results are about the MSE, they should be a summary value. In addition, based on Figures 3, 4, and 6, the performance of the proposed method is not so satisfying in some cases when the standard deviation of measurement error is low.


**Q5 Detailed Comments To The Authors:**

Assumption 5 has some problems. No correlation is not equivalent to independence when the random variables do not necessarily follow the normal distribution. The statement that M is independent of X and \Delta N cannot not be true under model assumption (7).

There are many typo errors. For example, in formula 25, the range of j in x_j is 1 to s_1, not 1 to n. Please check all the typos very carefully.

In the experiments, the method is called MerrorKIV, in other sections, it is called MEKIV. They are not consistent.

The number of simulations in the experiments is too small.

**Q7 Justification For Your Score:**


I arrive at this overall assessment based on the main strengths,  weaknesses and my understanding of this paper. I think the strengths of this paper a bit outweigh its weaknesses.

**Q9 Complying With Reviewing Instructions:**

1: Yes.

---

### Official Review · Reviewer_86pu · 2022-04-12

**Q2(1) Originality/Novelty:** 3
**Q2(2) Significance/Impact:** 3
**Q2(3) Correctness/Technical Quality:** 3
**Q2(6) Clarity Of Writing:** 3
**Q6 Overall Score:** 7
**Q8 Confidence In Your Score:** 4

**Q1 Summary And Contributions:**

The work is focused on identifying causal effects from observed data when the treatment is corrupted by additive error. The authors proposed a kernel-based instrumental variable estimator and showed that the causal effect E[Y|do(X)] can be correctly estimated under certain assumptions. The authors showed that the estimated causal effect works better with their approach in their experiments.

**Q2 Assessment Of The Paper:**

More detailed information regarding each of these aspects is given below:

**Q2(4) Quality Of Experiments (Optional):**

2: Fair: The experimental evaluation is weak: important baselines are missing, or the results do not adequately support the main claims.

**Q2(5) Reproducibility:**

2: Fair: Key resources (e.g., proofs, code, data) are unavailable but key details (e.g., proof sketches, experimental setup) are sufficiently well-described for an expert to confidently reproduce the main results.

**Q3 Main Strengths:**

Estimation of the causal effect is an important topic, especially when there exists unobserved confounding and the treatment is corrupted by additive error. The paper tackles a non-trivial task and proposes a novel method to estimate the causal structure under their setting.

The theoretical contributions of this paper are interesting, and the experiments seem promising.

This paper is written and well-organized.

**Q4 Main Weakness:**

Lack of experiment regarding the usefulness of their method in a real-world dataset.

**Q5 Detailed Comments To The Authors:**

Regarding identifiability: can you explain a bit more about your assumptions? In your setting, is the E[Y|do(x)] identifiable only under Assumptions 1-2 and 4-6? It seems that the reason for introducing Assumptions 3 and 7-10 is to use the kernel method. I think it would be helpful if the authors first discuss the identification of their model, and then provide their approach to estimate the causal effect.

How can we know there exist measurement error in treatment X in practice? What happens if there exists measurement error in outcome Y. Can you give more discussion about them?

It would be helpful if the authors could show the usefulness of their method in a real-world dataset, e.g., the example of "industrialization level" discussed in the introduction.


**Q7 Justification For Your Score:**

Their setting is interesting and is a non-trivial task.

The proposed algorithm is both new and relevant to the community.

**Q9 Complying With Reviewing Instructions:**

1: Yes.

---

### Official Review · Reviewer_G8rJ · 2022-04-13

**Q2(1) Originality/Novelty:** 3
**Q2(2) Significance/Impact:** 2
**Q2(3) Correctness/Technical Quality:** 3
**Q2(6) Clarity Of Writing:** 4
**Q6 Overall Score:** 7
**Q8 Confidence In Your Score:** 4

**Q1 Summary And Contributions:**

This paper presents a method for estimating causal effects in an instrumental variable model in which the treatment is measured with error.

**Q2 Assessment Of The Paper:**

More detailed information regarding each of these aspects is given below:

**Q2(4) Quality Of Experiments (Optional):**

3: Good: The experimental evaluation is adequate, and the results convincingly support the main claims.

**Q2(5) Reproducibility:**

3: Good: Key resources (e.g., proofs, code, data) are available and key details (e.g., proofs, experimental setup) are sufficiently well-described for competent researchers to confidently reproduce the main results.

**Q3 Main Strengths:**

See below.

**Q4 Main Weakness:**

See below.

**Q5 Detailed Comments To The Authors:**

Disclaimer: My expertise lies in causal inference and measurement error and not in kernel methods, so my review focused on these areas.

This work addresses measurement error which is an important and oft overlooked (at least in the machine learning community) issue in causal inference. I found the paper well-written and clear. The main results are presented with sufficient context to fully understand them and the assumptions are clearly stated and explained. The authors should be commended for explicitly calling out assumptions they consider strong. Insofar as I have concerns regarding the paper, they lie in whether or not the specific model considered here has broad applicability. I think the simulation experiments are well done, but would have like to have seen an example of a real problem that the authors argue matches the necessary assumptions. Additionally, I would have liked to have seen some discussion and evaluation of sensitivity to these modeling choices. In particular, what happens is M and N are only approximately independent given X? Similarly, how important in practice is the $X \perp \Delta N$ assumption? Finally, in addition to the discussion of the strength of these assumptions, some discussion regarding why they are necessary would also be helpful. All-in-all I think this is a well-written, technically sound paper that should be accepted. Below are a few additional minor comments/questions:

1. Section 4.4: Can you expand on why this results in better computational efficiency? It seems that a latent variable model that uses a simple parametric form for P(X | Z, M, N) would be computationally efficient, if potentially biased. Do you actually mean sample efficiency (i.e., by estimating the lower dimensional parameter)?

2. In Fig. 4, I think a bias variance break down would be more informative than MSE. In particular, if I understood correctly, you proved consistency, but not unbiasedness. While, I would expect the gap between MEKIV and the oracle method to be largely due to additional variance and the gap between MEKIV and the two estimators using M and N to be due to additional bias (in fact, I might also expect the bias estimators to have similar variance to the oracle estimator), it would be good to confirm these intuitions.

Typos:

1. Page 4, right col: form --> from

2. Eq. 15: Missing $d\nu$.

3. Page 7, right col: mu --> \mu and prcedure --> procedure

4. Section 5: MerrorKIV --> MEKIV

5. Page 8, left col: An data --> A data

**Q7 Justification For Your Score:**

Technically sound, clearly written paper presenting a novel approach to an important problem.

**Q9 Complying With Reviewing Instructions:**

1: Yes.

---

### Decision · Program_Chairs · 2022-05-15

**Decision:**

Accept (Oral)

**Comment:**

Meta Review: This paper presents a non-parametric kernel-based approach for instrumental variable estimation in the presence of measurement error on the causal variable.
This is a clear accept.
There is a strong consensus among the reviewers that this is a very good paper.

Some prior work that should perhaps be included is:
Vansteelandt S, Babanezhad M, Goetghebeur E. Correcting Instrumental Variables Estimators for Systematic Measurement Error. Stat Sin. 2009;19:1223-1246.
https://www.ncbi.nlm.nih.gov/pmc/articles/PMC2743431/